# New Treatment Options in Metastatic Pancreatic Cancer

**DOI:** 10.3390/cancers15082327

**Published:** 2023-04-17

**Authors:** Marta Fudalej, Daria Kwaśniewska, Paweł Nurzyński, Anna Badowska-Kozakiewicz, Dominika Mękal, Aleksandra Czerw, Katarzyna Sygit, Andrzej Deptała

**Affiliations:** 1Department of Oncology Propaedeutics, Medical University of Warsaw, 01-445 Warsaw, Poland; marta.fudalej@wum.edu.pl (M.F.);; 2Department of Oncology, Central Clinical Hospital of the Ministry of Interior and Administration, 02-507 Warsaw, Poland; 3Department of Health Economics and Medical Law, Medical University of Warsaw, 01-445 Warsaw, Poland; 4Department of Economic and System Analyses, National Institute of Public Health NIH-National Research Institute, 00-791 Warsaw, Poland; 5Faculty of Health Sciences, Calisia University, 62-800 Kalisz, Poland

**Keywords:** pancreatic cancer, KRAS, oncology, targeted therapy, immunotherapy

## Abstract

**Simple Summary:**

The poor prognosis of pancreatic cancer (PC) is associated with several factors, such as diagnosis at an advanced stage, early distant metastases, and remarkable resistance to most conventional treatment options. The pathogenesis of PC seems to be significantly more complicated than originally assumed. To develop effective treatment schemes prolonging patient survival, a multidirectional approach encompassing different aspects of the cancer is needed. Particular directions have been established; however, further studies bringing them all together and connecting the strengths of each therapy are needed. This review aims to provide an overview of new or emerging therapeutic strategies for the more effective management of metastatic PC.

**Abstract:**

Pancreatic cancer (PC) is the seventh leading cause of cancer death across the world. Poor prognosis of PC is associated with several factors, such as diagnosis at an advanced stage, early distant metastases, and remarkable resistance to most conventional treatment options. The pathogenesis of PC seems to be significantly more complicated than originally assumed, and findings in other solid tumours cannot be extrapolated to this malignancy. To develop effective treatment schemes prolonging patient survival, a multidirectional approach encompassing different aspects of the cancer is needed. Particular directions have been established; however, further studies bringing them all together and connecting the strengths of each therapy are needed. This review summarises the current literature and provides an overview of new or emerging therapeutic strategies for the more effective management of metastatic PC.

## 1. Introduction

Pancreatic cancer (PC) is the seventh leading cause of cancer death across the world, with an incidence of 2.5% and a mortality of 4.5% [1]. Its global burden has more than doubled over the past 25 years [2]. PC mainly affects older patients, with a median age of 70 at diagnosis. Only ~10% of patients are diagnosed before the age of 55 [3]. PC is asymptomatic until the disease reaches its advanced stage, which is directly associated with a low survival rate. Indeed, no screening program has been introduced to improve prognosis through early diagnosis [4]. Although the exact causes of PC development are still insufficiently explored, several risk factors have been established, such as cigarette smoking, positive family history and genetics, diabetes mellitus, obesity, dietary factors, alcohol use, and lack of physical activity [5]. PC types can be divided into two large cohorts—exocrine and neuroendocrine PC. Over 90% of PC cases develop in the exocrine tissue and, of these, pancreatic ductal adenocarcinoma (PDAC) is the most frequent histological subtype [6]. The American Joint Committee on Cancer (AJCC) introduced tumour, lymph node, and metastasis classifications to stratify patients and assess prognosis [7]. Multidisciplinary treatment is the most common choice for patients with diagnosed PC and encompasses surgery, chemotherapy, chemoradiotherapy, and supportive care [8,9,10]. Unfortunately, in most cases, patients are diagnosed at an advanced or metastatic stage of the disease. The most common sites of distant metastases are the liver (90%), lymph nodes (25%), lung (25%), peritoneum (20%), and bones (10–15%). In these cases, palliative chemotherapy is the only treatment option, and nearly all patients eventually relapse and require second-line options with limited choice [11,12,13,14,15].

Translational discoveries have shown that PC is highly heterogeneous at the molecular and cellular levels, and this is partially responsible for the high levels of resistance to treatment [16]. In the last decade, we have witnessed an increased interest in PC biology that has produced a substantial increment in knowledge of pancreatic cancer progression. Now the major challenge is to translate this acquired knowledge into better patient outcomes [17]. One of the initiating events in 70–95% of PDAC is an activating point mutation in the Kirsten rat sarcoma viral oncogene homologue (KRAS) oncogene. [18,19]. Multiple studies suggest that multigene profiling, including the determination of the KRAS status should be considered as a routine diagnostic workup (Figure 1). Patients with PDAC with wild-type KRAS represent a distinct subgroup who may benefit from further comprehensive molecular profiling to improve their outcomes [20]. This review summarises the current literature and provides an overview of new or emerging therapeutic strategies for the more effective management of metastatic PC.

## 2. KRAS

The Kirsten rat sarcoma viral oncogene homologue (KRAS) encodes the KRAS protein—a small guanosine triphosphatase (GTPase) acting as a molecular switch for different cellular processes and cycling between an active guanosine triphosphate (GTP)-bound and an inactive guanosine diphosphate (GDP)-bound state to regulate signal transduction [16,26]. The KRAS protein connects cell membrane growth factor receptors with transcription factors and intracellular signalling pathways, which results in cellular growth, proliferation, and survival [16]. KRAS mutation is an early event present in stage 1 pancreatic intraepithelial neoplasia [27]. An activating point mutation of the KRAS oncogene is the initiating event in most PDAC cases [18,19]; thus, the Ras signalling pathway might become a prime target for inhibitor development and present an effective treatment modality for PDAC [27]. Various research studies have been conducted so far; nevertheless, two issues have arisen: first, the long-standing reputation of KRAS as an undruggable molecule; the second—the existence of escape mechanisms despite KRAS inactivation [16]. As various targeted therapies have failed to elicit sufficient clinical benefits, KRAS seems to be an elusive target in PDAC. The structure of the KRAS protein lacks deep hydrophobic pockets, and its high affinity to GTP prevents the development of effective KRAS inhibitors [28]. Various highly selective and potent inhibitors against kinases in the Ras signalling network have been developed.

In PDAC, the most mutations in the KRAS gene occur in codon 12. There are five common mutations at codon 12, and these mutations account for >90% of KRAS mutations; G12D is the most common mutation (51%), followed by G12V (30%), G12R (15%), and G12C/S (2% each). It is now clear that KRAS mutations differ in their sensitivity to targeted drugs; even those in codon 12 are not all equal. A MEK1/2 inhibitor—selumetinib—had some activity against heavily pretreated KRASG12R-mutant PDAC and offered a median PFS of 3 months and median OS of 9 months. However, the results of the study suggest that alternative strategies beyond MEK inhibition with a single agent are required [21,27]. Perhaps a combination of a MEK 1/2 inhibitor plus chemotherapy could be more effective. Cobimetinib plus gemcitabine were applied to patients suffering from the metastatic KRASG12R-mutant PDAC that progressed after FOLFIRINOX and gemcitabine/nab-paclitaxel chemotherapy. In a small group of six patients who were treated with such a combination, one partial response and five stable disease responses were achieved with a median progression-free survival of 6 months and overall survival of 8 months [29].

The other G12D and G12V mutations have poor prognoses. Nevertheless, in the case of KRAS Gly12Cys (KRASG12C) mutations, there are early indications that inhibitors (e.g., sotorasib and adagrasib) might be effective in solid tumours with a KRASG12C mutation, especially among patients with non-small-cell lung cancer [30]. Sotorasib specifically and irreversibly inhibits KRASG12C by trapping it in the inactive GDP-bound state via interaction with the P2 pocket of the switch II region, which is present only in the inactive GDP-bound conformation of KRAS [26]. The data from the newest phase I/II CodeBreaK100 trial confirmed that sotorasib demonstrated clinically meaningful anticancer activity with a 21.1% objective response rate and an 84.2% disease control rate among patients with pretreated KRASG12C-mutated advanced pancreatic cancer [22,23]. KRASG12C is an uncommon mutation found in approximately 2% of pancreatic tumours [27]; however, if this approach proves effective, further efforts should be put to inhibit more common mutations. On the basis of sotorasib anticancer activity, another clinical study is planned to determine the safety, tolerability, and efficacy of sotorasib in combination with chemotherapy (liposomal irinotecan + 5 fluorouracil + leucovorin or gemcitabine + nab-paclitaxel) for patients with advanced KRASp.G12C-mutant pancreatic cancer with the progression of the disease after first-line treatment (NCT05251038).

Due to difficulties in targeting the KRAS mutation itself directly or through downstream effector pathways, other approaches have been proposed and encompass cancer vaccines [31,32].

Up to 12% of PC do not harbour the KRAS mutation [33,34]. Distinctly different molecular compositions of KRAS wild type (KRAS WT) tumours suggest potentially different molecular pathogenesis mechanisms of this type of PC [20]. Whole-genome sequencing improved the definition of the genetic landscape of KRAS WT PC and presented the occurrence of several genetic alterations, which can become potential targets for therapies [35]. Multiple studies conducted on KRAS WT tumours identified targetable oncogene fusions such as ALK, BRAF, FGFR2, MET, NRG1, NTRK1, NTRK3, RAF1, ROS1, EGFR, ERBB4, FGFR3, and RET. To provide new and meaningful treatment advances, prospective studies are needed to identify the optimal method of targeted treatment for each oncogene fusion [36]. For example, NRG1 rearrangement confers susceptibility to ERBB inhibitors and anti-epidermal growth factor receptor (EGFR) antibodies. So far, single trials have demonstrated the effectiveness of an HER-family kinase inhibitor—afatinib—in patients with an oncogenic NRG1 fusion [24,37]. As another example, scientists reported a case series of four patients diagnosed with PC presenting ALK-fusions who were treated with an ALK-inhibitor with promising results [38]. In terms of BRAF-driven PC cases, they are supposed to be sensitive either to BRAF inhibitors, MEK inhibitors, or combined therapeutical strategies [39]. One of the ongoing clinical trials is analysing the combination of a BRAF inhibitor—encorafenib—and a MEK inhibitor—binimetinib—for the treatment of PC in patients with the BRAF V600E mutation (NCT04390243). Recently, nimotuzumab—a humanised monoclonal antibody against EGFR—combined with gemcitabine improved median overall survival by 2.5 months over the gemcitabine plus placebo (11.9 vs. 8.5, respectively; *p* = 0.025) in a prospective randomised-controlled, double-blinded multicentre phase III pivotal trial for locally advanced or metastatic pancreatic cancer, with a decrease of 50% in mortality risk (HR = 0.50). Subgroup analyses revealed even more survival benefits in a group of patients without biliary obstruction (11.9 vs. 8.5 months; HR = 0.54; *p* = 0.037) and without surgical procedures (15.8 vs. 6.0 months; HR = 0.40) [40].

Further clinical studies might provide an accurate method for targeting KRAS WT fusions and other molecular changes and, as a result, offer more effective therapeutic strategies [36].

## 3. Others

### 3.1. Germline Mutations—BRCA

Regular DNA damage occurs due to both endo- and exogenous stressors; thus, cells have evolved a complex DNA damage response (DDR). A key hallmark of carcinogenesis—genomic instability—arises owing to defects in the DDR with or without increased replication stress. Defects in the DDR provide targetable vulnerabilities relatively specific to cancer cells that can be exploited for clinical benefit with the use of DDR inhibitors [41]. The most accurately studied class of DDR inhibitors are the inhibitors of the poly(ADP-ribose) polymerase (PARP-inhibitors) [42]. PARP inhibitors (PARPis), for example, olaparib, target cancer cells with a homologous recombination repair (HRR) deficiency. The most prominent target gene is the breast cancer gene (BRCA). PARPis trap DNA–PARP-1 complexes and disrupt the PARP-1 catalytic cycle, leading to replication fork destabilisation and consequent double-strand breaks. For tumour cells with BRCA mutations, HRR loss results in cell death [43,44]. PDAC was reported to present a close relationship with BRCA gene mutations, thus patients might benefit from treatment with PARPis [45]. In the Pancreas Cancer Olaparib Ongoing (POLO) trial, olaparib, as a maintenance treatment, improved progression-free survival (PFS) compared with the placebo after platinum-based induction chemotherapy in patients with PDAC and germline BRCA1/2 mutations (7.4 months and 3.8 months, respectively; HR 0.53; *p* = 0.004); nevertheless, no significant difference in overall survival (OS) between the trial groups was observed [46]. Significantly prolonged PFS should translate to a better health-related quality of life; thus, olaparib was approved in 2019 as a maintenance therapy for germline BRCA-mutated metastatic PDAC patients, despite the lack of statistically significant OS improvement [47]. At the annual Gastrointestinal Cancers Symposium in January 2021, the investigators reported an update on the study outcomes and presented facts potentially influencing survival. They stated that the study was inadequately powered to detect a difference in OS between the two groups. Moreover, OS might have been biased since patients in the placebo group received multiple subsequent lines of therapies upon disease progression and after stopping the study medication. Additionally, 26% of patients in the placebo arm received olaparib after disease progression [48]. On the other hand, the randomised phase II study SWOG S1513 evaluated the safety and efficacy of second-line treatment with other PARPis—veliparib and mFOLFIRI versus FOLFIRI alone (control) —for PDAC patients; however, the authors concluded that neither PFS nor OS was improved with the addition of veliparib [49].

Some preclinical studies proved that PARPis modulate the immune microenvironment by, among others, increasing programmed cell death ligand 1 (PD-L1) expression [50]. Going further, several clinical studies conducted on solid tumours demonstrated the preliminary efficacy of the PARP and immune checkpoint inhibitor combination. Based upon these data, the randomised phase II SWOG S2001 trial aims to evaluate the PFS of metastatic pancreatic cancer patients with germline BRCA1 or BRCA2 mutations treated with olaparib + pembrolizumab (anti-PD-L1 antibody) compared with olaparib alone as maintenance therapy [51]. Furthermore, Lundy et al. [52] (2022) have recently described a case of metastatic pancreatic adenocarcinoma harbouring a germline BRCA1 mutation and a high tumour mutation burden that demonstrated an excellent response to initial platinum-based chemotherapy followed by a complete radiologic response to maintenance immunotherapy with pembrolizumab and PARP inhibition. Based on the fact that the addition of PARPis to the immune checkpoint blockade could complement the clinical benefit of immune checkpoint inhibition, another precision medicine phase II study is being conducted. The trial assesses the efficacy and safety of the combination of olaparib, durvalumab (anti-PDL1 antibody) and tremelimumab (anti-CTLA4 antibody) in patients with several types of solid cancers, including pancreatic cancer, with at least one mutation in homologous repair genes [53]. Some of the ongoing clinical trials on PARP inhibitors in PC treatment are presented in Table 1.

### 3.2. Tumour Stroma

The PDAC tumour microenvironment (TME) consists of carcinoma-associated fibroblasts (CAFs), pancreatic stellate cells (PSCs), pericytes, neurons, endothelial cells, infiltrating immune cells, and extracellular matrix (ECM) proteins [54,55]. A desmoplastic reaction to the tumour is a histopathological hallmark of PDAC, both in primary tumours and metastases [56]. Desmoplasia is known to be responsible for creating a mechanical barrier around the PDAC tumour cells. As a result, it prevents appropriate vascularisation, limits exposure to chemotherapy, and causes poor immune cell infiltration [54]. Several attempts were made to disrupt TME as a potential therapeutic target for PC. One of them focused on hyaluronan, a major constituent of the stromal ECM. Researchers used human recombinant PH20 hyaluronidase (PEGPH20) for hyaluronan depletion. Despite the first promising results in which the addition of PEGPH20 enhanced the effects of gemcitabine plus nab-paclitaxel as shown by improved PFS [57], further results in PC treatment were rather disappointing [58,59]. Another approach aimed to target specific signalling pathways responsible for the development of the tumour stroma. Hedgehog (HH) signalling plays a major role in TME development with its involvement in myofibroblast differentiation and the induction of stroma-derived growth-promoting molecules. The expression of its primary ligand, Sonic Hedgehog (SHH), is an early event in pancreatic carcinogenesis and correlates with KRAS mutation [60]. Pre-clinical studies proved that HH pathway inhibition altered the fibroblast composition and immune infiltration in the PC microenvironment and thus might become a meaningful therapeutic target [61]; nevertheless, clinical trials with HH inhibitors turned out to have unsuccessful results. In the phase II study, the addition of the HH inhibitor vismodegib to chemotherapy did not improve efficacy in patients with newly diagnosed metastatic PC [62]. These results were in line with previous ones, in which the addition of vismodegib to gemcitabine in an unselected cohort also did not improve PFS and OS in patients with metastatic PC [63]. A dense fibroblast-rich stromal matrix covering the tumour nest in the PC limits the access of therapeutic agents to the tumour cells [64]. To overcome this issue, cyclic RGD (Arginine-Glycine-Aspartic acid) peptide ligand-decorated nanocarriers were devised to target abundantly expressed ⍺vβ3 and ⍺vβ5 integrin receptors of tumour vascular endothelial cells in the pancreatic tumour to deliver the therapeutic plasmid DNA [65].

Several contradictions were observed between preclinical and clinical responses and across several clinical trials focused on stromal desmoplasia. The tumour-promoting role of desmoplasia is well established; however, accumulating evidence demonstrates that desmoplasia is not entirely tumour-promoting. It is a physical barrier limiting drug exposure, but it also exhibits protective effects in restraining tumour growth and further progression [54,59]. The aforementioned failures imply also that targeting desmoplasia alone is not sufficient.

### 3.3. Angiogenesis

Different anti-angiogenic drugs have been approved by the Food and Drug Administration (FDA) for the treatment of various malignancies, including colorectal cancer [66], renal cell carcinoma [67], and ovarian cancer [68]. Yet, in terms of PC cancer treatments, multiple clinical trials of anti-angiogenic agents have been carried out with disappointing results. PC is characterised by an excessive deposition of the stromal matrix, which affects angiogenesis; thus, further exploration of stromal depletion should be conducted. Other possible mechanisms for the poor efficacy of anti-angiogenic therapies in PC encompass vessel co-option, vasculogenic mimicry, and vasculogenesis. They represent alternative and compensatory mechanisms of tumour growth and progression and thus may play a key role in resistance [69]. Anti-angiogenic therapies are based on the theory that tumours are unable to grow without the proper formation of new blood vessels [70]; nevertheless, while this treatment hinders the blood supply to the tumour, it also diminishes drug delivery at the same time [71,72]. Some studies have focused on vascular normalisation to enable proper drug delivery. Jacobetz et al. (2013), in the genetically engineered mouse model of PDAC, demonstrated that the enzymatic depletion of hyaluronan induced re-expansion of tumour blood vessels, which results in an increased intratumoural delivery of chemotherapeutics and, as a result, prolonged the survival of the mouse model [73]. Chauhan et al. (2013) proved that the angiotensin inhibitor losartan reduces stromal collagen and hyaluronan production, which results in increased vascular perfusion and drug delivery and potentiates chemotherapy in PC models [74]. Losartan, along with other agents, is now under investigation in clinical trials concerning both its role in blood flow improvement and transforming growth factor beta (TGF-β) function suppression in PC. The suppression of the TGF-β function is believed to stop cancer cells from becoming resistant to chemotherapy [25] (clinical trials: NCT05077800, NCT04106856, NCT03563248).

The vascular endothelial growth factor (VEGF), fibroblast growth factor (FGF), and platelet-derived growth factor (PDGF), together with their receptors, are highly expressed in PC; however, single-target anti-angiogenic agents have been studied for combined therapy in PC with limited success [75]. Resistance to VEGF-directed therapy may derive from signalling from other angiokinases. To overcome this therapeutic failure, a triple angiokinase inhibitor targeting VEGFR1/2/3, FGFR1/2/3 and PDGFRα/β signalling—nintedanib—was explored in the preclinical models of pancreatic cancer [76]. Although, in vitro, nintedanib neither presented anti-proliferative effects nor sensitised tumours cells to chemotherapy, in vivo, it blunted primary tumour growth and metastasis, reduced microvessel density and fibroblast activation, induced hypoxia, but did not promote an epithelial–mesenchymal transition in multiple preclinical models of PC. In the subsequent study, conducted on experimental PDAC, nintedanib presented strong antitumor activity both as a single agent and in the combination with gemcitabine [77]. Based on the above findings, nintedanib is currently in a clinical trial as a combined therapy with gemcitabine plus nab-paclitaxel for advanced PC (NCT02902484).

Another direction is associated with gene delivery strategies, which have successfully delayed PC progression via an anti-angiogenesis approach. Generally, strategies for anti-angiogenesis gene therapy might be divided into two categories: (1) the delivery of genes encoding endogenous angiogenesis inhibitors or their receptors; or (2) blockage of excessive genes encoding growth factors or their receptors. As an example, in xenografted pancreatic tumours, the exogenous delivery of a nanocarrier loaded with a plasmid DNA encoding soluble VEGF-1 (or soluble fms-like tyrosine kinase-1), a potent antiangiogenic protein, captured VEGF, exerting an antiangiogenic effect [78,79]. In another study, conditionally replicative adenovirus expressing human endostatin presented a special ability to duplicate and kill PC cells both in in vitro and in vivo models [80].

### 3.4. Immunotherapy

So far, immunotherapy has proven effective in many malignancies, including lung cancer [81], renal cell carcinoma [82], and melanoma [83]; however, the incorporation of this therapy to the standard treatment method of PC has met with numerous obstacles [84,85]. The key contributors to these failures are a low tumour mutational burden and immunosuppressive TME—PC is described as an immunologically “cold” tumour [54,86]. Extensive analyses of PC genomic datasets showed that only a small subset of PC is immunologically active [54]. The TME of PC is primarily characterised by the poor infiltration of effector T cells and prominent myeloid inflammation [84]. Nevertheless, recent studies proved that theories about the poor immunogenicity of PC might be slightly oversimplified. The low immunogenicity and antigenicity of PDAC are regulated by complex mechanisms influencing the interplay of myeloid, lymphoid, and stromal cellular compartments; thus, to overcome resistance, equally sophisticated treatment strategies should be developed [87].

Monotherapy with immune checkpoint inhibitors targeting CTLA-4 and the programmed cell death protein-1 (PD-1)/programmed cell death ligand-1 (PD-L1)) has mostly failed to elicit efficacy in patients with PC [88,89]. The reasons for this failure are multifactorial—first of all, only a very small subset (~1%) of PC patients present a high burden of microsatellite instability (MSI-high). Moreover, PC has low baseline PD-1+ T-cell infiltration into the tumour [84]. Nonetheless, for patients with mismatch repair deficient or MSI-high PC, the FDA has approved the anti-PD-1 immune checkpoint inhibitor—pembrolizumab [90]. Other approaches were proposed to combine immune checkpoint inhibition with different therapies. Some trials combined an immune checkpoint blockade with chemotherapy based on an immunogenic aspect of chemotherapy; however, most of them are phase I studies, with no conclusive beneficial evidence reported [14,91,92,93]. Radiotherapy was also proved to be immunogenic [94]; thus, new strategies aim at the re-introduction of radiotherapy in the treatment of PC. Mouse tumour models demonstrated improved survival and tumour volume reduction under a combination of radiotherapy and PD-L1 blockade compared with a single modality. Moreover, after radiotherapy, the expression of PD-L1 in tumour cells was elevated [95]. McCarthy et al. (2021) described a case report of near complete pathologic response to pembrolizumab and radiotherapy in a patient with locally advanced PDAC [96]. The recent phase II study revealed that the combination of stereotactic body radiotherapy plus pembrolizumab and trametinib (mitogen-activated protein kinase (MEK) inhibitor) could be a novel treatment option for patients with locally recurrent pancreatic cancer after surgery [97].

Chimeric antigen receptor (CAR)-T cell therapy represents an emerging therapeutic option for PC. This therapy utilises genetically engineered T cells redirected to specific cancer-associated antigens to elicit potent cytotoxic activity [98]. Current CAR-T cell therapies fail to improve survival in PC patients. The main barriers are created by the distinct TME; thus, recent efforts have focused on targeting TME along with CAR T cell therapy [99]. Solid tumours have additional challenges compared with haematological diseases, and the success of CAR T-cell therapy in PC requires a more sophisticated approach. For example, the co-administration of checkpoint-blocking antibodies or cell-intrinsic PD-1 resistance mechanisms might further improve CAR T-cell efficacy [100].

Efforts to broaden the impact of immunotherapy in PDAC are focusing on two major approaches. The first approach is to stimulate the anti-tumour T cell responses by multi-targeted strategies. A second approach examines immune-based strategies to condition tumours for enhanced responsiveness to chemotherapy [92]. All the examples described above prove that immunotherapy might become effective in PC treatment, but definitely as a part of a multi-agent strategy rather than monotherapy. Different therapeutic strategies, encompassing immunotherapy, are presented in Figure 2. 

### 3.5. Cancer Vaccines

Various vaccine-based studies have been conducted in PC. Therapeutic cancer vaccines encompass whole-cell, dendritic cell, DNA, and peptide vaccines. They provoke the presentation of immunogenic cancer antigens (tumour cells, tumour-related proteins, genes expressing tumour antigens) to the immune system and result in cancer antigen-specific cytotoxic T lymphocyte activation and a subsequent anti-cancer immune response [101]. One of the well-known vaccines is GVAX, an allogenic whole-cell cancer vaccine generated from a PDAC cell line genetically modified to express the granulocyte-macrophage colony-stimulating factor [102]. A phase II study from 2015 showed that PC patients with previously treated disease achieved better OS with the combination of GVAX with cyclophosphamide (Cy) and CRS-207 (live, attenuated Listeria monocytogenes expressing mesothelin) compared with historical OS achieved with chemotherapy alone. However, a subsequent three-arm study from 2019 analysing GVAX/Cy + CRS-207 (arm 1), CRS-207 alone (arm 2), and standard chemotherapy (arm3) did not meet its primary efficacy endpoints and demonstrated that the combination of Cy/GVAX + CRS-207 did not improve survival over chemotherapy [103].

PD-1/PD-L1 blockade with vaccine therapy was proved to facilitate the infiltration of effector T cells in PC [104]. Vaccination is suspected to improve tumour immune recognition in metastatic PC and increase the response to PD-1/PD-L1. A combination of vaccines with immune checkpoint blockades may improve the outcomes of PC patients [105]. The latest reports from ongoing clinical trials confirmed that combining GVAX with the dual immune-targeting of PD-1 blockade and CD137 agonism may enhance disease-free survival (DFS) in resectable PDAC patients treated in peri-operative and post-adjuvant settings (NCT02451982) [106]. Currently, researchers are also conducting a phase II trial of GVAX/Cy combined with nivolumab (anti-PD-1) and stereotactic body radiation therapy followed by definitive resection for patients with borderline resectable PDAC (NCT03161379).

Considering other vaccines, in the phase II study, the peptide cocktail vaccine OCV-C01 combined with gemcitabine showed a median DFS of 15.8 months, which was an improvement compared with gemcitabine alone (a DFS of 12.0 months) [73]. IMM-101 is a systemic immune modulator comprising heat-killed whole-cell Mycobacterium obuense [107]. Neves et al. (2015) [108] first published a case report describing a patient with metastatic PC who underwent a synchronous resection of the primary tumour and metastatic site after multimodality neoadjuvant therapy with gemcitabine, nab-paclitaxel and immunotherapy backbone with IMM-101, as well as consolidation chemoradiotherapy. As a result, the pathological examination of specimens showed a complete response at both sites, and the patient remained alive for four years from initial diagnosis, with continued maintenance immunotherapy. Afterwards, in a randomised, open-label, phase II study, IMM-101 combined with gemcitabine among 101 untreated patients with advanced PC improved OS (7.2 months vs. 5.6 months with gemcitabine alone) and PFS (4.4 vs. 2.3 months). These results should warrant further evaluation in an adequately powered confirmatory study [109]. Currently, the exact benefit of introducing cancer vaccines to PC treatment is still under investigation. For example, neoantigen vaccines are studied both as neoadjuvant (NCT05111353) and adjuvant therapeutic strategies (NCT03956056, NCT04810910).

### 3.6. Nanocarriers

As previously described, pancreatic tumours are characterised by immunosuppressive and desmoplastic TME, which creates major challenges in developing effective treatment. Nanocarriers present the potential to revolutionise cancer therapy. Advances in protein engineering have contributed to novel nanoscale targeting approaches that may improve the prognosis of oncological patients [110]. Latest advances in nanotechnology have enabled the development of multiple nanocarrier-based formulations. These improve both drug delivery and immunotherapy-based approaches for PC [111]. Nanoparticle-based strategies for delivering anticancer drugs and biologics substantially minimise undesirable systemic spread. Effective nanomedicines should circulate in the blood while avoiding unspecific interactions with blood components and reticuloendothelial systems (RES) [112]. The non-specific capture of nanocarriers by RES organs, particularly the liver, causes a substantial decrease in the delivery efficiency of nanomedicines to target tissues and impairs their pharmacokinetic properties [113,114]. To avoid the adsorption of opsonin proteins, which facilitate nanomedicine recognition and elimination by RES, nanomedicines are modified with biocompatible materials, such as polyethylene glycol (PEG) [115,116,117]. Another concern is associated with factors affecting drug delivery and penetration in the stroma of pancreatic cancer. 30 nm sized nanocarriers were proved to effectively penetrate the poorly permeable pancreatic tumours to achieve antitumor activity compared to 70 nm sized nanocarriers [118].

In 2015, the FDA approved liposomal irinotecan in combination with 5-FU and leucovorin for clinical use to treat patients with metastatic PDAC who had previously been treated with gemcitabine-based chemotherapy. Patients treated with all three drugs gained, on average, 2 months of survival and had a 3.1-month delay in tumour growth [119]. Nevertheless, due to exposure to blood proteins, liposomal formulations lose selectivity, which increases their toxic effect [120]. Recent studies in PDAC animal models demonstrated that the mesoporous silica nanoparticle platform might improve irinotecan loading, efficacy, and safety [121]. Another example of a widely used clinical-stage nanomedicine for PC therapy is the nanoparticle albumin-bound paclitaxel (nab-paclitaxel). The nanoparticle albumin-bound platform allows for formulating hydrophobic drugs while largely mitigating the need to use toxic excipients. As a result, the drug is better tolerated and can be used at higher doses with faster administration [122]. Recent studies suggest that the use of chitosan-based nanoformulation as a carrier of gemcitabine and 5-FU improves the therapeutic effect and decreases toxicity in PC. Chitosan is a linear polysaccharide which is able to affect some characteristics of drugs, such as half-life, toxicity, time of circulation, and release profile [123].

### 3.7. Microbiome

The human microbiome is defined as the microorganisms that reside in the human body, such as bacteria, viruses, fungi, protozoa, and their genomes. Numerous studies proved that gut microbiota influences the host immune response to a tumour and affect the response to cancer treatment. Currently, the association between locally resident or intratumour microbiota and carcinogenesis and outcomes of cancer therapies is the focus of microbiome research [124]. In terms of PC, Geller et al. (2017, 2018) demonstrated that analysed PDAC tumour samples harboured Gammaproteobacteria, which are capable of metabolising gemcitabine to the inactive form of 2′,2′-difluorodeoxyuridine using cytidine deaminase [125,126]. In another study, analysing PDAC patients receiving adjuvant gemcitabine, authors proved that patients without Klebsiella pneumoniae (belonging to Gammaproteobacteria) in bile culture had better progression-free survival (PFS) than those with K. pneumoniae. Additionally, treatment with quinolones improved median overall survival [127]. The aforementioned studies suggest that microbial dysbiosis might induce gemcitabine resistance, and appropriate antibiotic therapy may reverse this resistance and improve patient outcomes.

As previously discussed, immunotherapy in PC was found to be of limited effectiveness. The microbiota may alter the tumour microenvironment and immune cell infiltration in PC and potentially impact the efficacy of immunotherapy [128]. In the PDAC orthotopic mouse model, antibiotic-mediated microbial ablation enhanced antitumour immunity and increased susceptibility to checkpoint-targeted immunotherapy by upregulating PD-1 expression on effector T cells [129]. Thus, the combination of checkpoint-directed immunotherapy with specific microbiota ablation might emerge as a potential treatment strategy for PC [130]. Microbiome-based immunotherapy for PC should be tailored to specific bacterial taxa and immune cells. Microbial ablation studies demonstrated that the immunosuppressive microenvironment in PDAC is mediated by microbial Toll-like receptor (TLR) ligation; nevertheless, the exact functions and immunogenic properties of specific microbiota are still under investigation [131].

A microbiota-derived approach should be taken into account, and numerous clinical trials evaluating the effect of the microbiome on PC are currently ongoing. A prospective translational tissue collection trial aims, among others, to establish molecular reasons for drug resistance and to investigate particular microorganisms colonising PC individuals and detect their impact on patient outcomes (NCT03840460). Another study is attempting to provide a new theoretical basis in which the gut microbiota regulates the subtype and anti-tumour effect of anti-mesothelin CAR-T therapy in PC (NCT04203459). The trial is based on the hypothesis established in other malignancies that gut microbiome modulation carries the potential for enhancing CAR-T cell responses [132]. The newest pilot study is designed to modulate the microbiome with ciprofloxacin and metronidazole to further enable the efficacy of neoadjuvant checkpoint-based immunotherapy with pembrolizumab following mFOLFIRINOX chemotherapy in surgically resectable PDAC (NCT05462496). Preclinical studies proved that the alteration of the microbiome modulates TME and the immune system, which affects the efficacy of chemotherapy and immune-targeted therapies in PC [133]. Drugs for PC with different mechanisms of action are summarised in Table 2. 

## 4. Conclusions

Developing effective treatment regimens to prolong the survival of patients with metastatic pancreatic cancer requires a multi-pronged approach to different aspects of cancer. The individual lineages described in this review have been established, and patients with metastatic PC now have more therapeutic strategies to choose from. However, the final results of many studies have yet to be published, so further clinical trials are needed to bring them all together and combine the strengths of each therapy.

## Figures and Tables

**Figure 1 cancers-15-02327-f001:**
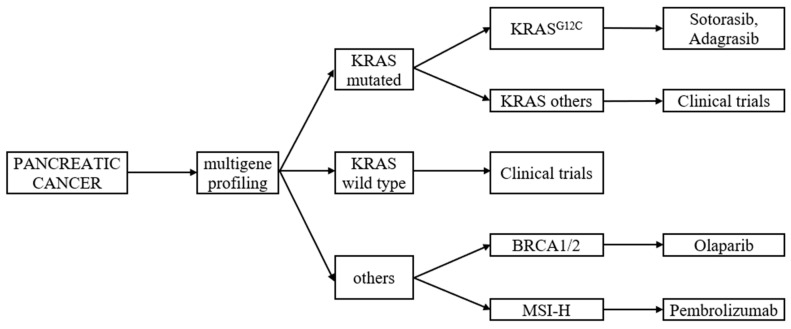
Multigene profiling of metastatic pancreatic cancer, with regard to KRAS mutation. Based on the literature review of [20,21,22,23,24,25].

**Figure 2 cancers-15-02327-f002:**
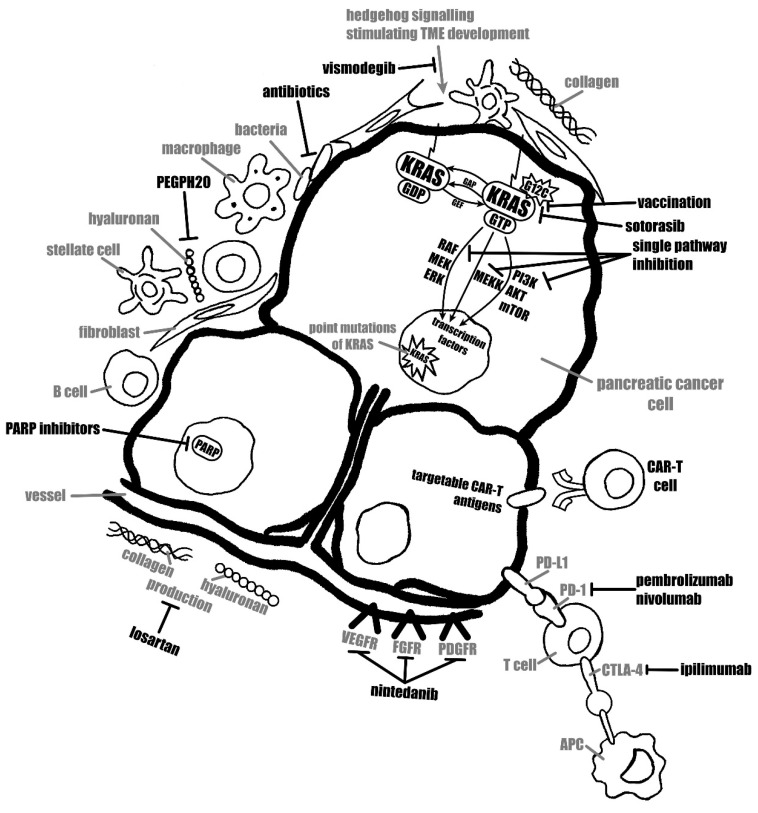
Scheme representing different therapeutic strategies.

**Table 1 cancers-15-02327-t001:** Ongoing clinical trials on PARP inhibitors in pancreatic cancer treatment (data derived from https://clinicaltrials.gov/ (accessed on 12 November 2022)).

Identifier	Phase	Mutations	Drug (or Combination)	Neoplasm
NCT04753879	Phase II	No specific genetic targets	Olaparib + pembrolizumab following multi-agent, low-dose chemotherapy with gemcitabine, nab-paclitaxel, capecitabine, cisplatin, and irinotecan (GAX-CI)	Untreated metastatic PC
NCT02498613	Phase II	No specific genetic targets	Olaparib + cediranib maleate	Metastatic/unresectable PC (among others)
NCT05411094	Phase I	No specific genetic targets	Olaparib + durvalumab + radiation therapy	Locally advanced, unresectable PC
NCT04858334	Phase II	BRCA1/2 or PALB2	Olaparib following the completion of surgery and chemotherapy	Resectable PC
NCT04409002	Phase II	No specific genetic targets	Niraparib + dostarlimab + radiation therapy	Metastatic PDAC
NCT03553004	Phase II	Genes involved in DNA repair	Niraparib	Metastatic PC
NCT04493060	Phase II	BRCA1/2 and PALB2	Niraparib + dostarlimab	Metastatic PDAC
NCT03140670	Phase II	BRCA1/2 or PALB2	Rucaparib	Locally advanced/metastatic PDAC (that has not progressed on platinum-based therapy)
NCT03337087	Phase II	BRCA1/2 or PALB2	Rucaparib + liposomal irinotecan + fluorouracil + leucovorin calcium	Metastatic PC (among others)
NCT04550494	Phase II	Genes involved in DNA damage response	Talazoparib	Locally advanced/metastatic PC (among others)

**Table 2 cancers-15-02327-t002:** Drugs approved by FDA for pancreatic cancer.

Drug	Mechanism of Action
Capecitabine	converts to its only active metabolite, fluorouracil, by thymidine phosphorylase [134]
Erlotinib	inhibits the intracellular phosphorylation of tyrosine kinase associated with the epidermal growth factor receptor [135]
Everolimus	inhibits the mTOR (mammalian target of rapamycin) serine/threonine kinase signal transduction pathway [136]
5-fluorouracil	inhibits thymidylate synthase and incorporation of its metabolites into RNA and DNA [137]
Gemcitabine	converts into active triphosphorylated nucleotides interfering with DNA synthesis and targeting ribonucleotide reductase [138]
Liposomal irinotecan	binds reversibly to the topoisomerase I–DNA complex and prevents repair of single-strand breaks [139]
Nab-paclitaxel	targets microtubules and causes mitotic arrest at G2/M phase [140]
Olaparib	inhibits the poly (ADP-ribose) polymerase [46]
Sunitinib	inhibits cellular signalling by targeting multiple receptor tyrosine kinases [141]

## Data Availability

The data presented in this study are available in this article.

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
