# Peer review of "New Treatment Options in Metastatic Pancreatic Cancer"

_cancers, 2023, doi:10.3390/cancers15082327_

Round 1

Reviewer 1 Report

The manuscript's structure is sound; the topic is reviewed in detail, summarising the current literature and evidence, and future directions are critically discussed. 

Reviewer 2 Report

The authors intended to review current treatment options for patients diagnosed with pancreatic cancer. In general the review is well written and illustrates the ongoing dilemma regarding the oncological outcome. 

Nevertheless there are still some concerns that I would like to raise. 

Major comments and findings:

  1. As the authors correctly stated, systemic relapse is a dilemma in PDAC. Solely systemic treatment options are addressed by this review. Surgical treatment remains the only potentially curative treatment. This is not addressed at all and novel strategies are not described. However, local tumor control is an ongoing dilemma which is also worth mentioning. As stated in the Introduction section the authors have mentioned the operative technique for pancreatic head cancer.
    1. However, resectability criteria are not only stratified by the arterial vasculature, but also by the distal superior mesenteric vein; please change this in line 56
    2. Local relapse in terms of local recurrence is present in 80% of all patients, this was not addressed in line 68
    3. By incorporating the circumferential resection margin into pathological evaluation, the underestimated local tumor burden was addressed and it was started to be understood. (dorsal resection margin and medial resection margin evenly at risk for R1 resections). 
    4. The mesopancreas was implemented for oncological resection, to address the issue with the dorsal resection margin, this has to be addressed for new treatment strategies  
    1. Studies have shown, that by incorporating the mesopancreas during resection, the local recurrence rates dropped. 

We encourage the authors to also discuss the dilemma with local tumor control and novel surgical treatment strategies in the manuscript in order to complete the review for all medical fields. Alternatively, the title may be changed to advances in conservative treatment...

We recommend the following Papers for citation:

  1. Esposito I, Kleeff J, Bergmann F, Reiser C, Herpel E, Friess H, Schirmacher P, Büchler MW. Most pancreatic cancer resections are R1 resections. Ann Surg Oncol. 2008 Jun;15(6):1651-60. doi: 10.1245/s10434-008-9839-8. Epub 2008 Mar 20. PMID: 18351300.
  2. Inoue Y, Saiura A, Yoshioka R, Ono Y, Takahashi M, Arita J, Takahashi Y, Koga R. Pancreatoduodenectomy With Systematic Mesopancreas Dissection Using a Supracolic Anterior Artery-first Approach. Ann Surg. 2015 Dec;262(6):1092-101. doi: 10.1097/SLA.0000000000001065. PMID: 25587814.

Reviewer 3 Report

Andrzej Deptała and team reviewed a topic entitled “New treatment options in pancreatic cancer.” The group mainly focused on emerging therapeutic strategies for managing pancreatic cancer, one of the hard-to-treat cancers. The manuscript is well-framed and well-presented and is worth publishing in the MDPI Cancres. However, the reviewer recommends a detailed revision addressing the following issues carefully to reach broader audiences and readers of different disciplines before considering a possible publication. 

1.  The reviewer recommends that the authors discuss their Future perspectives on the therapeutic management of Pancreatic cancer.

2.    The reviewer encourages the authors to list the drugs and their working mechanisms approved by different authorities, such as USFDA. For example, Everolimus (Afinitor) is an mTOR inhibitor approved for Pancreatic cancer (2011) by USFDA. 

3.    Enormous progress has been devoted to devising various nanoparticles for the effective and targeted delivery of drugs and biologics to the pancreas cancer site. Adding a section that describes nanocarrier-enabled pancreatic cancer therapeutic strategies would add value to the review. 

4.    The reviewer recommends that the authors discuss the challenges and potential problems of different strategies of pancreatic cancer treatment methods. For example, the systemic distribution of small-molecule anticancer drugs upon systemic administration routes results in chemotoxicity (Nat Nanotechnol 2007;2(12):751-60). Nanoparticle-based strategies for delivering anticancer drugs and biologics substantially minimized undesirable systemic spread (Chem Rev 2018;118(14):6844-6892). However, the nonspecific capture of nanocarriers by the reticuloendothelial system organs, particularly the liver, causes a substantial decrease in the delivery efficiency of nanomedicines to the target tissues(Nat Mater 2016;15(11):1212-1221, and Sci Adv 2020;6(26):eabb8133)

5.    The size of nanocarriers is one of the significant factors affecting drug delivery and penetration in the stroma of pancreatic cancer. 30-nm-sized nanocarriers effectively penetrated the poorly permeable pancreatic tumors to achieve antitumor activity compared to 70-nm-sized nanocarriers (Nat Nanotechnol 2011;6(12):815-23).

6.    Gene delivery strategies have successfully delayed pancreatic cancer progression via an anti-angiogenesis approach. I recommend that the authors discuss such emerging strategies in the relevant section of the manuscript. For example, exogenous delivery of nanocarrier loaded with a plasmid DNA encoding soluble vascular endothelial growth factor receptor-1 (or soluble fms-like tyrosine kinase-1), a potent antiangiogenic protein, captured vascular endothelial growth factor, exerting an antiangiogenic effect in xenografted pancreatic tumors (J Control Release 2015;209:77-87).

7.    A dense fibroblast-rich stromal matrix covering the tumor nest in the pancreatic tumors limits the access of therapeutic agents to the tumor cells (Cancer Sci 2009;100(1):173-80)To overcome this issue, cyclic RGD (Arginine-Glycine-Aspartic acid) peptide ligand-decorated nanocarriers were devised to target abundantly expressed vβ3 and vβ5 integrin receptors of tumor vascular endothelial cells in the pancreatic tumor to deliver the therapeutic plasmid DNA (Biomaterials 2014;35(20):5359-5368). 

Round 2

Reviewer 2 Report

This review is now better focused and reflects current treatment options in advanced metastasized PDAC

Reviewer 3 Report

Accepted in present form